# Preparedness for the Dengue Epidemic: Vaccine as a Viable Approach

**DOI:** 10.3390/vaccines10111940

**Published:** 2022-11-17

**Authors:** Md. Zeyaullah, Khursheed Muzammil, Abdullah M. AlShahrani, Nida Khan, Irfan Ahmad, Md. Shane Alam, Razi Ahmad, Wajihul H. Khan

**Affiliations:** 1Department of Basic Medical Science, College of Applied Medical Sciences, Khamis Mushayt Campus, King Khalid University (KKU), Abha 62561, Saudi Arabia; 2Department of Public Health, College of Applied Medical Sciences, Khamis Mushayt Campus, King Khalid University (KKU), Abha 62561, Saudi Arabia; 3Department of Chemical Engineering, Indian Institute of Technology Delhi, New Delhi 110016, India; 4Department of Clinical Laboratory Sciences, College of Applied Medical Sciences, King Khalid University (KKU), Abha 62561, Saudi Arabia; 5Department of Medical Laboratory Technology, College of Applied Medical Sciences, Jazan University, Jazan 45142, Saudi Arabia; 6Department of Chemistry, Indian Institute of Technology Delhi, Hauz Khas, New Delhi 110016, India; 7Department of Microbiology, All India Institute of Medical Sciences Delhi, New Delhi 110029, India

**Keywords:** dengue, *Aedes* mosquito, clinical presentations, diagnosis, dengue vaccines

## Abstract

Dengue fever is one of the significant fatal mosquito-borne viral diseases and is considered to be a worldwide problem. *Aedes* mosquito is responsible for transmitting various serotypes of dengue viruses to humans. Dengue incidence has developed prominently throughout the world in the last ten years. The exact number of dengue cases is underestimated, whereas plenty of cases are misdiagnosed as alternative febrile sicknesses. There is an estimation that about 390 million dengue cases occur annually. Dengue fever encompasses a wide range of clinical presentations, usually with undefinable clinical progression and outcome. The diagnosis of dengue depends on serology tests, molecular diagnostic methods, and antigen detection tests. The therapeutic approach relies completely on supplemental drugs, which is far from the real approach. Vaccines for dengue disease are in various stages of development. The commercial formulation Dengvaxia (CYD-TDV) is accessible and developed by Sanofi Pasteur. The vaccine candidate Dengvaxia was inefficient in liberating a stabilized immune reaction toward different serotypes (1–4) of dengue fever. Numerous promising vaccine candidates are now being developed in preclinical and clinical stages even though different serotypes of DENV exist that worsen the situation for a vaccine to be equally effective for all serotypes. Thus, the development of an efficient dengue fever vaccine candidate requires time. Effective dengue fever management can be a multidisciplinary challenge, involving international cooperation from diverse perspectives and expertise to resolve this global concern.

## 1. Introduction

Dengue virus (DENV) infection is the foremost expeditiously infectious mosquito-borne viral disease worldwide, which may result in dengue fever (DF). The worst part of dengue fever is the many severe outcomes, such as dengue hemorrhagic fever and dengue shock syndrome. Dengue fever is an acute disease, which places a significant socioeconomic and disease load on several tropical and subtropical regions and is the foremost frequent flavi/arboviral infection worldwide [1,2]. The increasing rate of dengue fever is higher than any other infectious disease, with a 400% increase over only 13 years (2000–2013), as reported in “The Global Burden of Disease” [3]. It has been documented that dengue has been neglected, but the statistics reveal huge investments in developing a vaccine and innovative control measures for the vector in the past couple of years [4]. Recently, we observed how quickly and cheaply the COVID-19 vaccine was produced, resulting in the immunization of a considerable proportion of the world’s population. This will provide a hint that vaccination for other viral diseases, such as dengue, may be attainable in future [5,6,7]. This review article is an attempt to highlight the current dengue-related trends in general, including the work that was put into developing the vaccine, as well as the clinical results. The prevention and treatment of dengue fever depend on several strategies, one of which is vaccination, the most important part in disease management.

## 2. Characteristic Features of Dengue Virus and Its Serotype

DENV is a mosquito-borne virus belonging to the *Flavivirus* genus of the *Flaviviridae* family. It includes several clinically important human pathogens, as well as mosquito-borne viruses, such as yellow fever virus (YFV), Japanese encephalitis virus (JEV), West Nile virus (WNV), and also tick-borne viruses, such as tick-borne encephalitis virus (TBEV). Infection with the dengue virus poses a threat to endemic regions for many reasons, the important ones being the presence of four serotypes, vector control problem, inadequate treatment modality, and vaccine unavailability [8,9]. DENV is an enveloped virus, with a virion size of 50 nm, and comprises one strand of positive-sense RNA encoding three structural proteins, such as the capsid (C), membrane (M), and envelope (E), and seven non-structural (NS) proteins, such as NS1, NS2A, NS2B, NS3, NS4A, NS4B, and NS5, with a genome of approximately 11 kilobases [10]. The nucleocapsid, which contains the viral genome and C proteins, is found within the virus. The nucleocapsid is surrounded by a membrane known as the viral envelope, a lipid bilayer derived from the host. The E and M proteins are embedded in the viral envelope and span the lipid bilayer. The membrane protein (prM/M) and envelope (E) proteins present 180 copies within the well-defined glycoprotein shell. [11]. The envelope (E) protein consists of three discrete domains, particularly DI, DII, and DIII [12]. In addition, DIII is thought to have varied type-specific neutralizing epitopes. Due to the concern as an immunogen, the E protein is the most vital part of DENV vaccines [13]. Dengue virus has four serotypes, namely DENV-1, DENV-2, DENV-3, and DENV-4, based on the interaction with the antibodies found in human blood serum. The virus and the serotype are depicted in Figure 1. Although the four dengue viruses are similar and share around 65% of their genomes, even one serotype can exhibit considerable genetic diversity. Despite these variations, all dengue serotype infections result in the same sickness and set of clinical signs. The development of a dengue vaccine is greatly hampered by variations in the serotypes.

## 3. Incidence/Prevalence of Dengue Disease

Dengue prevalence has risen dramatically worldwide in the last ten years. Many cases are asymptomatic or mild and self-managed, so the exact numbers of dengue cases are not reported. The febrile nature of many similar diseases results in the misdiagnosis of dengue fever [14]. The prevalence of dengue is high, especially within the Asia-Pacific region and the Americas. Four antigenically distinct but genetically homogeneous serotypes (DENV1-4) are circulating in these areas. However, some DF patients develop an astringent syndrome known as the Dengue hemorrhagic fever (DHF), in which patients may exhibit hematomas with marked thrombocytopenia or astronomically low platelet counts [15].

Severe signs and symptoms are apparent when dengue infection results in the Dengue shock syndrome (DSS), which is a disease like the dengue hemorrhagic fever that manifests a coagulation disorder, increased fragility of vessels, increased plasma leakage, fluid loss due to raised capillary permeability and altogether progressing to a hypovolemic shock state, and an increased chance of failure of multiple organs of the body [16]. Plasma leakage lasts for about two days and is the clinical hallmark of dengue hemorrhagic fever, leading to decreased circulatory volume [17]. After dengue infection, bleeding is commonly noticed, particularly in the case of DHF/DSS. Most of the ensuing deaths are due to symptoms associated with DHF/DSS [18,19]. For dengue, there is no antiviral treatment modality; however, vaccines have been licensed in many dengue-prone nations. Dengue is beginning to lead as one of the life-threatening emerging vector-borne diseases [20,21].

About 390 million dengue cases are estimated every year [1]. Some researchers also estimated that approximately 3.9 billion individuals are at risk of dengue, particularly in 128 endemic nations [22], and Asia being the host of 70% of the dengue burden [1]. During the middle of the current decade, even in America, about 2.35 million dengue cases occurred. Out of these, about ten thousand cases were related to severe hemorrhagic fever. As a result, the WHO has prioritized the development of a vaccine in order to address this emerging issue; otherwise, dengue could become a big threat to global public health, adding a major contribution to the burden of disease at the global level [1,22].

## 4. Epidemiology and Global Expansion

Dengue is a disease that presents itself with varied clinical features. In some cases, the recovery from it is spontaneous, being non-lethal; however, in some instances, it takes a deadly form because of the leakage of plasma without hemorrhage or being associated with hemorrhage. The group proceeding from non-rigorous to astringent illness is hard to outline; however, this will be a crucial concern, since proper treatment might stop the illness from evolving into more rigorous clinical situations. Amendments to the epidemiology of dengue result in problems based on the present WHO classification. The clinical manifestations of DENV infection can vary from mild–acute indistinguishable febrile sickness to classical dengue fever (DF, duration → 2–7 days), dengue hemorrhagic fever (DHF, duration → after 3–5 days of fever), and dengue shock syndrome (DSS, duration → after 3–5 days of fever) based on the WHO 1997 dengue guidelines [23]. DF is an ingenious febrile sickness, which presents symptoms, such as headaches, leukopenia, rash, bone or joint and muscular pains. The clinical presentation includes high-grade fever, hemorrhage, hepatomegaly, and circulatory failure in severe cases [24]. The dengue guidelines for diagnosis, treatment, prevention, and control were updated by the WHO in 2009 and are shown in Figure 2 [25].

Some infected people might develop hypovolemic shock resulting from severe plasma leakage (Figure 3). Some chronic sicknesses have been held to be responsible for triggering dengue severity [24,26,27]. The geographical dissemination of each vector and dengue viruses (DENVs) has led to the global rejuvenation of epidemic DF and the emanation of DHF in the last few decades. The WHO has steered an updated guideline, classifying dengue infection into dengue and severe dengue [25,28].

Travelers play a significant role in the epidemiology of dengue worldwide, as infected travelers carry different serotypes and strains of dengue into endemic areas [29]. Travelers transmit dengue from developing tropical countries with fewer laboratories to developed nations with adequate facilities to identify dengue virus serotypes [30].

Among the most rapidly transmissible vector-borne diseases globally is dengue. Over the past five decades, there has been a 30-fold increase in the incidence and transmission of dengue to nations where it was not endemic. Surprisingly, this pattern has been noted as a shift from urban to rural areas in the recent decade. About 50 million cases of dengue have been reported, and nearly 2.5 billion human beings live in nations where dengue is said to be endemic [31]. The World Health Organization (WHO), within the World Health Assembly (WHA) resolution in 2002, emphasized a firm commitment to dengue [32], and later on, the WHA resolution in 2005, while revising the International Health Regulations (IHR), highlighted dengue as a global public health issue. About 70% of the at-risk global population belongs to SEAR, as well as to the Western Pacific region, and accounts for approximately 75% of the worldwide load of dengue [33]. The Asia-Pacific Strategy created for dengue (2008–2015) has been devised in deliberation with the representatives of different countries and development allies in response to the escalating threat from dengue, which is transmitted to non-endemic new areas and results in increased mortality during the initial phase of the epidemic [33]. The strategic set-up’s goal is to enable countries to invert the ascending flow of dengue by mobilizing their states to notice, characterize, and include outbreaks expeditiously and to prevent the dissemination to new areas. The dengue epidemic has disseminated to new areas and has increased in the areas of regions already affected since 2000. During 2003, India, Indonesia, Bangladesh, Myanmar, Sri Lanka, the Maldives, Thailand, and Timor-Leste witnessed many dengue infections. In 2004, Bhutan faced a dengue epidemic for the first time. In 2005, the WHO’s agency GOARN reacted responsibly to an epidemic of dengue with case fatality as high as 3.55% in Timor-Leste. For the first time, near the end of 2006, Nepal also witnessed dengue. There is no evidence of dengue cases in the Democratic People’s Republic of Korea. The SEAR countries have four different climatic zones, each with a varied potential for the transmission of dengue.

Dengue is one of the leading public health issues in Indonesia, Timor-Leste, Sri Lanka, Myanmar, and Thailand, as these nations have a wider circulation of the vector for dengue, i.e., *Aedes aegypti.* This is because these areas are located in the tropical monsoon and equatorial zone, and dengue, with its many serotypes, is the main cause of hospitalization and mortality among children both in rural and urban areas. Cyclic epidemics are recorded in the Maldives, Bangladesh, and India; a precondition for such epidemics is a transient dry and wet climate region with several virus serotypes spreading around [33]. Outbreaks of dengue were documented in Egypt around 1799 [34]. The outbreaks increased in frequency around 1985 in Sudan [35] and around 1991 in Djibouti [36]. Suspected dengue epidemic outbreaks were recorded in Pakistan, Sudan, Yemen, and Saudi Arabia in 2005–2006 [34]. In the mid-1990s, Pakistan faced its first outbreak of dengue fever. Again, in the middle of the first decade of the 21st century, DEN-3 outbreak along with hemorrhage was also reported for the first time.

Subsequently, the development of dengue disease with increased prevalence and rigor has been recorded in giant cities in Pakistan and as far north as the North-West Frontier Region in 2008. In Pakistan, dengue is included in the list of reportable diseases [37]. Since the first mortality in Jeddah (1993) due to DHF, the Kingdom of Saudi Arabia has faced three major epidemics of DEN-2 in 1994, 2006, and 2008. These were attributable to Jeddah being a Haj entry point for Muslims from all over the world as the largest port city in KSA from a commercial point of view and the topmost city in western KSA with the most bustling airport and huge numbers of passengers landing from endemic countries, such as Indonesia, Thailand, and Malaysia, i.e., countries in a region already affected by dengue [38,39,40,41].

## 5. Transmission of Dengue Virus/Fever

Different dengue virus serotypes transmit the dengue disease to human beings by the already infected female vector bite, i.e., *Aedes* mosquito, mainly *Aedes aegypti*. This vector could be a tropical and subtropical species extensively dispensed worldwide, predominantly between latitudes 35° N and 35° S. These geographical peripheries accord more or less with a winter isotherm of 10 °C. *Aedes aegypti* was found as far north as 45° N; however, such incursions emanated throughout warmer months, and the mosquitoes did not survive winters. *Aedes aegypti* is relatively uncommon above 1000 m because of lower temperatures [22,40].

In artificially collected waters, the *Aedes* breed, mostly female *Aedes aegypti,* could be killed in or around the households wherever they appear as adults. This denotes that individuals, not mosquitoes, expeditiously move the dengue virus within and between communities. In addition to *Aedes aegypti*, dengue epidemics are often attributed to *Aedes albopictus*, *Aedes polynesiensis*, and several other species of the *Aedes scutellaris* complex. All such species have specific ecology, behavior, as well as distribution. The eggs will remain viable for several months without water [40,41].

After 4–10 days of incubation, the infection caused by any of the four virus serotypes (1–4) can induce a wide range of illnesses, even though most infections are asymptomatic or subclinical. First-time infection is said to induce immunity for the rest of one’s life for that particular serotype. The infected individuals are forfended from clinical illness with a distinct serotype within 2–3 months of the first infection but without an extended cross-protective immunity. The risk factors of an individual influence the rigor of the disease and include secondary infection, age, ethnicity, and possibly chronic diseases (bronchial asthma, sickle cell anemia, and diabetes mellitus) [40,41].

Humoral and cellular immune responses are analyzed to furnish virus clearance through the generation of neutralizing antibodies and also the activation of CD^4+^ and CD^8+^ T lymphocytes. Moreover, the innate host defense could limit infection from the virus. After infection, the serotype-specific and cross-reactive antibodies and CD^4+^ and CD^8+^ T cells remain accessible for many years. Plasma leakage, hemoconcentration, and abnormalities in homeostasis represent severe dengue [40].

## 6. Preventive Measures for Dengue Fever

Dengue fever is a significant public health issue in the Western Pacific Region. Epidemics have persisted in the region in abundance since the last severe pandemic in 1998. The lack of reporting is among the foremost important obstacles to dengue prevention and control.

The prevention and control of dengue infection are enforced through the Bi-regional Dengue Strategy (2008–2015) of the WHO’s South-East Asia and Western Pacific regions. This comprises six elements, i.e., Surveillance of dengue, Management of cases, Response to outbreak, Management of integrated vector, Dengue’s social mobilization and communication, and Research on dengue (the combined effect of both formative and operational research) [40,41].

In 2008, this strategy was already supported by the WHO [42]. Information regarding dengue fever among travelers, utilizing sentinel surveillance, could also be shared expeditiously to make the international community aware of the onset of epidemics in endemic areas where there is no surveillance or reporting of dengue fever, as well as the geographic escalation of virus serotypes and genotypes to new areas, which increases the prospect of severe dengue.

Dengue cases reported worldwide show a cyclical difference between soaring epidemic and non-epidemic years. Dengue is usually identifiable within mass outbreaks. In addition, there is also a seasonality to the dengue fever, with outbreaks ensuing in different phases of the year. This seasonality is marked by the highest level of disease spread, driven by features of the host, the vector, and therefore, the agent [40,41].

## 7. Diagnostic Methods for Dengue

The biomarkers that are used for the diagnosis of dengue infection embrace the virus itself (isolating the virus in culture or mosquitoes or viral genomic RNA detection), the viral products (capture and secreted NS1 protein detection), or the host immunologic response to viral infection (through measuring virus-specific immunoglobulin G (IgG) and immunoglobulin M (IgM)). A recent study presented the timing of the occurrence and the extent of these biomarkers in both primary and secondary dengue infection [25].

The diagnosis of dengue depends on serologic assays, antigen detection, and molecular diagnostics (Table 1 shows all the available diagnostic methods in brief). The most vital and widely used tests in serology are hemagglutination inhibition, enzyme-linked immunosorbent assay (ELISA), and immunofluorescence antibody assays [43]; however, they are restricted by inconsistent sensitivity and specificity because of cross-reactivity with other flavivirus infections and flavivirus vaccines [44]. In acute dengue infection, serum IgM becomes positive within four to five days and is ensured by IgG development after the seventh day of sickness. The sensitivity of economically feasible ELISAs in diagnosing acute dengue may vary from 60% to 90%, and specificity from 80% to 99% [43].

The specific tests for dengue disease embrace RNA detection procedures and non-structural (NS1) antigen tests. Polymerase chain reaction (PCR) tests allow the first association of dengue RNA throughout the viremic phase (often within the first five days of the appearance of the illness) [46,47,48]. PCRs have many important features, such as rapidity, the specificity of serotypes (including detection of synchronous infections by variant serotypes), faculty for determinable quantifications, and high sensitivity (92–98.5%) and specificity (92.4–100%) [46,47,48]. In a recent study, the NS1 antigen detection test has become extensively acquirable. The NS1 antigen detection test is very distinct for acute dengue disease (95–100%); however, its sensitivity is variable (65–85%) [49,50,51,52]. The high specificity of the NS1 antigen means that a negative result does not rule out the diagnosis of dengue; however, a positive test denotes a high likelihood of confirming the diagnosis. NS1 antigen testing is more perceptive when employed in the first three days after the appearance of fever in patients with primary infection, high-level viremia, DENV-1 infection, and in patients with DF compared with DHF and DSS [50,51,53].

The dynamics of the contents of viremia and NS1 antigenemia in serum differ in primary and secondary infection, beyond the various serotypes of dengue, and with severity of disease [54]. There is an increase in the diagnostic output of acute dengue (85–99%) when we combine NS1 antigen or PCR with the IgM/IgG assay in endemic communities and travelers [49,53,55]. In a study regarding the achievement of NS1 antigen in the case of travelers, the sensitivity of NS1 antigen detection was found to be the highest on days 6–7 after the onslaught of the sickness, which differed from NS1 antigen detection studies in the case of endemic communities, where sensitivity was found to be highest around day 3 after the appearance of fever [55,56].

It is troublesome to differentiate primary and secondary dengue infection by serology, but it is also attainable by selecting the ratio of IgM to IgG. On day 6 of the sickness, an IgM: IgG ratio of ≥1.78 is considered to be steady with a primary infection, while a ratio of <1.2 signifies a secondary infection [46,57]. IgG avidity assays in non-immunized persons can also be used to indicate primary versus secondary dengue infections. Unfortunately, a perfect method of diagnosis, which enables initial and fast diagnosis and is cheap, easy to operate, and has strong attainment, is not available to date [58,59].

## 8. Diagnostic Assays in Development

There are various latest approaches for a fast diagnosis of dengue disease, which are presently in progress. These advanced assays include micro/paper fluidics, isothermal PCR [60], in vivo micro-patches, electrochemical, and piezoelectric detection. These technologies are in the initial phase of progress, compelling perpetuated processing to create a practical explanation in the natural world. The most crucial objective for diagnosing dengue would be an assay that differentiates between primary and secondary dengue infection based on IgG and IgM capture and significant serotype-specific NS1 detection [60]. Arbovirus control is problematic because of the lack of effective vaccinations, antiviral medications, insecticide-resistant vectors, including *Aedes* species, and vector control measures that reduce the human–vector contact [61,62]. Developing innovative methods to detect and control arboviruses is a public health priority in this environment. Nanobiotechnology has been recognized as a future technology with extraordinary new benefits [63,64,65,66,67]. In vector control, nanoparticles could be used to develop new drugs with higher activity, decreased toxicity, and sustained release. Additionally, they could be used to develop new repellent formulations based on natural or synthetic compounds, as well as to develop biosensors that can quickly detect and diagnose mosquito-transmitted viral diseases [68,69,70,71]. Nanoparticles have been widely used for decades because of their small size (nanoscale), mobility, ability to multi-task, ability to adapt, increased solubility, ability to create personalized medicines, ability to find diseases early, and ability to prevent diseases [72,73,74,75,76]. Nanomedicine has been used successfully to improve treatment for a wide range of illnesses, such as neural, cancer, cardiopulmonary, and communicable diseases, such as HIV-1, HBV, influenza virus, and respiratory syncytial virus [77,78,79,80,81]. Nanobiotechnology could be used to treat arbovirus patients due to the absence of medications for viral infections.

## 9. Treatment of Dengue Fever

There are no adequate therapies for treatment nor a vaccine for the prevention of dengue disease, and the mortality from DHF/DSS continues to be elevated. The treatment has been mostly supportive, and to date, no licensed therapeutic drug is available. DENV non-structural protein 1 (NS1), which could be secreted in patients’ sera, has been used as a primary marker for the diagnosis of dengue infection for several years. The pathogenic roles of NS1 in dengue-induced hemorrhage and vascular leakage and the prospect of using NS1 as a therapeutic target and vaccine competitor are articulated in a most recent study [82].

## 10. Viral Life Cycle and Host Immune Response

DENV will transmit to humans via an infected mosquito. Numerous possible receptors have been proposed, even though the cell type and binding receptor are still not completely known. Among the cells susceptible to DENV infection are dendritic cells (DCs), endothelial cells, fibroblasts, keratinocytes, macrophages, mast cells, and monocytes [83]. Clathrin-mediated endocytosis will allow the virus to enter the cell by receptor attachment to the DENV E glycoprotein. The low pH of the endosomal compartment will change the conformation of the viral proteins, leading to membrane fusion and the eventual release of viral DNA into the cytoplasm. Replication is linked to the viral replication complex (RC), a cellular membrane structure carried on by the virus. The positive-sensed RNA genome may begin to be translated into a polyprotein by the host ribosome [84]. After viral and host proteases break down the polyprotein, the structural prM and E proteins assemble in the ER lumen. After protein translation and genome replication, the virus is assembled and transferred to the Golgi apparatus, where the host furin protease degrades the prM to produce the mature virion [84]. The mature, infectious virion is released via exocytosis.

When DENV is injected into the skin, the immune sentinels easily recognize it [85]. It has been demonstrated that DENV can be bound in vitro by proteoglycan, heparin sulphate, and glycosaminoglycans, which are often expressed on different mammalian cell types. Several immune defenders, including dendritic cells (DCs), Langerhans cells (LCs), macrophages, and mast cells (MCs), are present at the site of the mosquito bite in the skin [85,86]. Certain skin cells, including DCs and macrophages, are recognized as target cell types for DENV infection. Either productive or unsuccessful DENV infection can activate the innate immune responses against the virus by activating the pattern recognition receptors. Langerhans cells, DCs, macrophages, and mast cells, which are epidermis immune cells of the hematopoietic lineage, encounter DENV.

Additionally, they can start the production of new inflammatory mediators by either transcriptional activation or enzymatic activation (e.g., cytokines, such as TNF). Degranulation of MCs by DENV and inactivated DENV particles promotes oedema and the recruitment of cytotoxic cells. Plasma leakage, which causes endothelial damage and extravascular plasma leakage, is the defining feature of DHF and DSS (Figure 4).

## 11. Dengue Vaccine Development and Progress

Based on the approach related to the purified inactivated virus or live-attenuated virus, numerous vaccine candidates for dengue [87,88,89,90,91,92,93] are in the advanced stage of clinical trials. The present status of dengue vaccine development based on the live-attenuated or inactivated virus is given here (Table 2). To date, Sanofi Pasteur CYD-TDV has been approved under the name of Dengvaxia^®^ (Sanofi Pasteur, Lyon, France) [87]. Some other important vaccine competitors are in the progressive clinical trials, such as TV003/TV005 (NIH), TDEN (WRAIR/GSK), and TDV (Takeda/Inviragen) [88,89,90,91,92]. CYD-TDV, an authorized dengue vaccine, uses dengue virus 1–4 prM/E on its YF-17D backbone. TV005 and TV003 are identical to TV003, except the dosing levels are based on strains of wild-type virus having undergone genetic mutations to reduce the viral effects after its attenuation [93].

## 12. Dengue Vaccine Platform and Vaccine in a Developmental Phase

A total of four different dengue vaccines are now being developed—(a) live-attenuated virus vaccines, (b) inactivated virus vaccines, (c) recombinant subunit vaccines, (d) DNA vaccines [1]—and are presented in Figure 5. The immune response to these vaccines is schematically represented in Figure 6. These platforms and the vaccines developed upon them are summarized here and tabulated in Table 2.

### 12.1. Live-Attenuated DENV Vaccines

The live-attenuated virus vaccines, one of the five different kinds of dengue vaccines, have undergone the most extensive testing on humans and are therefore the most advanced. Live-attenuated vaccines are antigenic substances created from a living pathogen modified to be less virulent or avirulent. Once given, the viruses proliferate locally, triggering cell-mediated immune reactions and neutralizing antibodies against the four dengue virus serotypes. These vaccines show the advantages of delivering protective antigens while providing long-lasting immune protection [104]. A number of live-attenuated dengue vaccines have been created using recombinant DNA technology, of which CYD-TDV developed by Sanofi Pasteur, TAK-003 developed by Takeda, and TV003/TV005 developed by the National Institute of Allergy and Infectious Diseases (NIAID) are the three most advanced dengue vaccines. The following are examples of live-attenuated vaccines currently in different stages of development.

**Dengvaxia © (CYD-TDV):** The only licensed tetravalent live-attenuated dengue vaccine candidate is Dengvaxia, also known as CYD-TDV [94,105,106]. By replacing the YF17D prM and E regions with those of the four DENV serotypes, CYD-TDV employs the yellow fever 17D (YF17D) vaccine strain as its base. The overall vaccination effectiveness (VE) was between 56.5% and 60.8%. The protection against DENV1 and DENV2 was about 40–50%, whereas the protection against DENV3 and DENV4 was over 70% [94,105]. Its use is permitted with severe limitations on the serostatus and age of the recipients [107]. It has high efficacy in preventing dengue infections brought on by DENV serotypes 1–4 and is safe for seropositive individuals who have already acquired dengue. The FDA has so far approved its use in people aged 9 to 16 who previously had dengue illness confirmed in a lab and who reside in endemic areas [108]. For those who are seronegative, however, the vaccine raises the risk of developing severe dengue when the person contracts the disease spontaneously about 3 years after vaccination. The formation of neutralizing antibodies against all four DENV serotypes is induced by vaccination in naïve subjects. This response is mostly driven by specific antibodies against one or a small number of serotypes, while cross-reactive antibodies primarily drive reactions against other serotypes.

Additionally, it generates T-cell responses against DENV structural antigens, which are serotype-specific and cross-reactive. Seronegative patients may therefore present a subclinical attenuated illness, which is “primary-like”. Depending on the serotypes, vaccination in this group also has varying immunological consequences. Seronegative people exhibit limited cross-protection compared to seropositive individuals who have cross-protection prompted by vaccination, which increases the chance of causing antibody-dependent enhancement (ADE) [109,110,111]. As a result, the WHO advises that only seropositive people receive this vaccine. The efficacy, immunogenicity, and side effects are summarized in Table 3.

**Tetravax (TV003/TV005):** In contrast, the viral particle structure, infectivity, and immunogenicity of Tetravax (TV003/TV005) are very different from those of CYD-TDV [88,95,112]. Researchers used structural gene prM/E chimerization and nucleotide deletions in the 3′ untranslated region (UTR) to lessen the pathogenicity of DENV. CYD had lower dengue virus type 2 resistance, a lower level of adaptive immune response, and a higher risk of viremia compared to TV003/TV005 [88]. The efficacy, immunogenicity, and side effects are summarized in Table 3.

**TAK-003:** A live-attenuated, chimeric, tetravalent dengue vaccine under the name of TAK-003 is referred to as DENVax [96]. It is still in phase III of clinical research at this time. A DENV2 strain with reduced virulence (PDK-53), which contains the prM/E components of all serotypes serves as the vaccine’s main component. Independently of the participant’s age or serostatus, the vaccine is immunogenic and well tolerated in numerous phase I and II clinical investigations. Similar to its predecessor Dengvaxia, TAK-003 demonstrated a DENV-serotype-dependent protective efficiency. However, TAK-003 had greater levels of DENV2-neutralizing antibodies and lower DENV3 and DENV4 protection rates; therefore, its safety profile is not well understood [113]. Although a prior clinical investigation found it activates CD^8+^ T cells specific for NS1, NS3, and NS5 in patients who have never had a DENV infection [114], this information was left out of the subsequent clinical studies. The efficacy, immunogenicity, and side effects are summarized in Table 3.

**TDEN F17/F19:** TDEN F17/F19 vaccines are an alternative to live-attenuated dengue vaccinations [97,115]. This vaccine was demonstrated to be a secure, well-tolerated, and immunogenic DENV vaccine candidate in a phase II trial. More than half of the infants/toddlers and all the children had antibody responses to all four DENV types one month after the second dose [116]. In these investigations, monovalent vaccinations were lyophilized and blended into a tetravalent vaccine at the point of administration. The efficacy, immunogenicity, and side effects are summarized in Table 3.

### 12.2. DENV DNA Vaccine

Nucleic acid vaccines have also been developed for dengue and now progressed to phase 1 clinical trials. The following are examples of DNA-based vaccines currently in different stages of development.

**D1ME100:** The pre-membrane (prM) and envelope (E) genes of the DENV-1 virus are included in the monovalent DNA plasmid vaccination known as D1ME100 (US Naval Medical Research Center) [101,117,118,119,120,121]. Kochel et al., found that mice injected with a monovalent DEN-2 DNA vaccination, which included the prM gene and a truncated E gene that encoded 92% of the envelope protein, developed anti-DEN-2 neutralizing antibodies and were protected from intracerebral viral challenge [120]. Subsequent studies evaluating the immunogenicity of DNA vaccines expressing truncated or full-length DEN-1 E genes showed that only constructions comprising 80% or the prM plus 100% of the E gene produced high neutralizing antibody levels. The prM 100% E construct also produced DEN-1-virus-like particles in HEK 293 cell culture supernatants [119]. Chimeric DNA vaccine constructions encoding antigens with epitopes from all four dengue serotypes have been evaluated in macaques to generate a vaccination that protects against all four dengue serotypes [121]. The macaques established NAb responses against all four dengue serotypes, but when challenged with DEN-1 or DEN-2 virus, they only partially protected against DEN-1 [121]. In a second investigation, the inoculation of rhesus monkeys with a tetravalent vaccine containing equal doses of monovalent DNA vaccines encoding the prM and E genes of dengue-1, -2, -3, or -4 viruses adjuvanted with Vaxfectin^®^ (Vical Inc., San Diego, CA, USA) produced NAb responses against DEN-1, -3, and -4 viruses. Compared to control animals, the animals were DEN-2 resistant (3). Vaxfectin^®^ increased vaccination immunogenicity.

The dengue virus serotype-1 (DENV1) vaccine design was used in Phase 1 clinical trial after the DEN-1 monovalent DNA vaccine was immunogenic in non-human primate models and protected against a live DEN-1 virus challenge (D1ME100). An open-label, dose-escalation, safety and immunogenicity experiment was conducted [101]. Healthy flavivirus-naïve adults received three needle-free Biojector^®^ 2000 intramuscular injections (0, 1, and 5 months) of a high-dose (5.0 mg) or low-dose (1.0 mg) DNA vaccination. The vaccination was harmless. The high-dose group had 41.6% anti-dengue NAb, while the low-dose group had none. The high-dose group had stronger T-cell responses (83% detectable) than the low-dose group (50% detectable) [117]. In a phase 1 dosage escalation research, a tetravalent vaccine formulated in Vaxfectin^®^ was tested for safety, tolerability, and immunogenicity (ClinicalTrials.gov NCT01502358).

**Tetravalent dengue DNA vaccine (TVDV):** TVDV was prepared by combining equal amounts of plasmid DNA vaccines that encode dengue 1, 2, 3, and 4 pre-membrane (prM) and envelope (E) genes cloned into VR1012. West Pac 74 prM and E genes are in the dengue-1 DNA vaccine. Ravi prakash et al. report the plasmid vaccine construction [119,121]. Dengue-3 [122] and dengue-4 DNA vaccines have low-passage Philippine strain prM and E genes. The dengue-2 DNA vaccine uses the prM and E genes of a low-passage Philippine strain with E’s C-terminal transmembrane and cytoplasmic domains replaced by lysosome-associated membrane protein 1 (LAMP-1). In mice, LAMP-1 replacement increased anti-dengue 2 neutralizing antibody responses [121,123,124].

### 12.3. Subunit DENV Vaccine

There are numerous recombinant subunit vaccine candidates currently being developed and mainly based on surface glycoprotein E. E protein is the main target of a neutralizing antibody and the best option for a subunit vaccine. Recombinant subunit vaccines still have difficulties in eliciting a strong immune response, necessitating an efficient adjuvant approach. The following is an example of a subunit vaccine currently in the developmental phase.

**V180 (DEN-80E):** V180 (DEN-80E), a potential tetravalent vaccine, is one of the most effective vaccine candidates that has successfully completed the phase I clinical trial. In V180, each DENV serotype N-terminal constituting 80% of the DENV envelope glycoproteins (E) (DEN-80E) was selected and successfully produced in Drosophila cells [125]. According to research, V180 formulations are typically well tolerated [100]. In a study, 18 final doses of DEN1-80E, DEN2-80E, DEN3-80E, and DEN4-80E vaccines were administered at 10 g each. This higher dosage was used because DEN4-80E showed less immunogenicity than the other serotypes, as shown in past investigations [100]. In a study, four recombinant, soluble dengue virus envelope glycoproteins constitute the V180 vaccine candidate, which has already undergone two clinical studies to test its immunogenicity and safety in *Flavivirus*-naïve volunteers (NCT01477580 and NCT0093642) [126].

### 12.4. Inactivated Dengue Vaccine

The C, M, E, and NS1 protein components of the inactivated dengue virus vaccine are antigens for eliciting immunity, although composite vaccines produce greater protectivity than single-type vaccinations. The following are examples of subunit vaccines currently in the developmental phase.

**TDEV-PIV:** TDEV-PIV, a tetravalent dengue virus purified inactivated vaccine, was developed by GlaxoSmithKline (GSK) and the Walter Reed Army Institute of Research (WRAIR) conducting phase I research trial. A synergistic formulation with a different live-attenuated candidate vaccination is also being evaluated in a phase II study (the prime-boost method). Prime boosting is administering a second dose of a different vaccine after the first in an effort to raise immunogenicity [127]. A study evaluated the immunogenicity and protective efficacy of TDENV PIV formulated with alum or an adjuvant system tested at three different dose levels in a 0, 1-month vaccination schedule in rhesus macaques. All adjuvanted formulations produced potent and enduring neutralizing antibody titers against all four dengue virus serotypes one month after the second dose. When the dengue serotype 1 and 2 virus strains were given 40 and 32 weeks following the second dose, respectively, most of the formulations tested inhibited viremia after a challenge. This study demonstrates that inactivated dengue vaccines show potential for further development, whether produced using alum or an adjuvant system [128]. The efficacy, immunogenicity, and side effects are summarized in Table 3.

**Table 3 vaccines-10-01940-t003:** Efficacy, immunogenicity, and side effects of some dengue vaccines in clinical trial.

Vaccine/Other Name	Efficacy	Serotype-Specific Efficacy	Immunogenicity	Comments/Side Effect	References
Dengvaxia^®^ (CYD-TDV) (ChimeriVax; ChimeriVax Tetravalent Dengue Vaccine; ChimeriVax-DEN1-4; ChimeriVax-Dengue; CYD dengue vaccine; CYD dengue vaccine—Sanofi; CYD-4444; CYD-5553; CYD-5555; CYD-TDV; CYD-TDV Dengue Vaccine; Dengue fever vaccine—Sanofi; Dengue Tetravalent Vaccine, Live—Sanofi; Dengue vaccine recombinant tetravalent—Sanofi; Dengue virus vaccine—Sanofi; Dengvaxia; Live-attenuated, dengue serotype 1, 2, 3, 4 virus vaccine; Tetravalent CYD 1, 2, 3, 4 dengue; Tetravalent dengue vaccine—Sanofi; TV-CYD)	25–59%	DENV-4 > DENV-3 > DENV-1 > DENV-2	CD^8+^ respond primarily to NS3, mainly DENV-4-neutralizing antibodies	Age restriction; increased risk of severe dengue in seronegative people; good efficacy and safety in seropositive persons; increases hospitalizations in vaccination recipients	[129,130,131]
DENVax/TAK003/TDV (DEN1-DEN2-DEN3-DEN4 vaccine—Takeda Pharmaceuticals USA; Dengue vaccine tetravalent—Takeda Pharmaceuticals USA; DENVax™; Needle-free dengue vaccine; QDENGA; TAK-003; TDV—Takeda Pharmaceuticals USA; Tetravalent dengue vaccine—Takeda Pharmaceuticals USA)	73.3–85.3%	DENV-2 > DENV-1 > DENV-4	Antibodies effective against all four serotypes	Children and teenagers tolerate it well; Independent of participant age or serostatus, immunogenic and well tolerated in several phase I and II clinical trials; Safety profile unknown	[89,132,133]
TetraVax-DV-V003/TV005 (Tetravalent live-attenuated dengue vaccine admixture TV 005; TetraVax-DV TV005; TetraVax-DV-Vaccine—Admixture TV005; TV-005)	Not yet released	DENV-4 > DENV-3 > DENV-1 > DENV-2	Rhesus macaque strong neutralizing antibodies	Single dose well tolerated; balanced immunological response; efficient with one dose administration; adverse response (mild rash)	[88,95,131,134]
TDEN LAV (Dengue vaccine live—GlaxoSmithKline; T-DEN F-19; T-DEN F17; TDENV-LAV F17; Tetravalent dengue live-attenuated virus formulation 17; WRAIR Live-Attenuated Tetravalent Dengue Vaccine)	Not yet released	Not yet released	Not yet released	Discontinued	[135]
TDENV PIV (Tetravalent purified inactivated dengue vaccine—GlaxoSmithKline/Walter Reed Army Institute)	Not yet released	Not yet released	Not yet released	No recent information, well tolerated and immunogenic in seropositive and naïve people. No reactivation risk and a healthy immunological balance	[128]

### 12.5. Commercially Available DENV Vaccine

Only the CYD-Tetravalent Dengue Vaccine (CYD-TDV) is commercially available so far and sold under the brand name Dengvaxia. CYD-TDV is a live-attenuated, recombinant tetravalent vaccine employing the attenuated YF virus 17D strain as the replication backbone. The vaccine was first approved in December 2015 on the basis of efficacy trials conducted in 10 highly dengue-endemic countries in Asia and Latin America, involving over 30000 children, and it has also been approved by regulatory authorities in ~20 countries. Phase 3 trials showed vaccine efficacy regardless of age, serostatus, and serotype, also showing a population-level benefit [136,137]. The interference manifested by asymmetric immunological responses against the mixtures of all four dengue vaccine viruses has been recognized as a possible reason for this varied vaccine performance [138].

In a most recent study, it was found that the efficacy of the vaccine against symptomatic dengue in seropositive vaccinees in the first 25 months was 76% (CI 63.9–84.0) based on the NS1 antibody assay, whereas in seronegative participants aged, it was 39% (CI 1 to 63). The long-term safety data up to 66 months, expressed as a hazard ratio against hospitalized dengue and severe dengue, are also discussed [139,140].

## 13. Use of CYD-TDV Vaccine and WHO Position

The live-attenuated dengue vaccine (CYD-TDV) has proven efficacy and safety in seropositive individuals. However, it carries a greater risk of severe dengue in people who experience their first natural dengue infection after vaccination (those who were seronegative at the time of vaccination). Countries should envisage introducing the CYD-TDV dengue vaccine as long as the minimization of risk among seronegative people is also assured. Vaccination should be considered as part of an integrated dengue prevention and control strategy. There is an ongoing need to adhere to different disease preventive measures, such as well-executed and sustained vector control. Whether vaccinated or not, people must seek prompt treatment if dengue-like symptoms occur [139,140].

The WHO has highlighted the criteria on the quality, safety, and efficacy of dengue tetravalent vaccinations based on the experiences gained throughout the development of prospective live-attenuated vaccines (live, attenuated). Additionally, the FDA claims that Dengvaxia, a dengue live-attenuated vaccine, induces dengue-specific immune responses against the four serotypes of dengue virus following injection, but the precise mechanism of protection is still unknown [108].

## 14. Summary and Conclusions

Over the last few decades, dengue has become the foremost important and widespread vector-borne virus infection affecting humans. Physicians and experts are still focusing on understanding dengue fever’s exact mechanism of pathogenesis. Further studies are urgently required to enhance our understanding of DHF pathogenesis, which is extremely important for developing effective therapeutics and vaccine strategies against DV infection.

The worldwide distribution of dengue is rapidly increasing and may continue until an efficient vaccine against dengue is developed. Various vaccine competitors for dengue include tetravalent, live-attenuated virus, purified inactivated virus, plasmid DNA, recombinant proteins subunit, and virus-vectored dengue vaccines. These vaccines are at preclinical and clinical trial stages based on different approaches. An intensive evaluation of cell-mediated and humoral immune response is vital for developing an efficacious vaccine for dengue primary prevention. Even after decades of research on dengue, a fully efficacious dengue vaccine candidate remains a challenge. The only recently authorized competitor in terms of a dengue vaccine, Dengvaxia, could not provide a balanced immune reaction against serotypes 1–4 of the dengue virus. This vaccine may provide long-lasting protection against all four serotypes of the dengue virus in a balanced manner and will also overcome the antibody-dependent enhancement (ADE) effect. With many dengue vaccine candidates in different stages of clinical trials, the approaching decade will see more numbers of licensed dengue-vaccine-supported recombinant protein approaches. An efficacious vaccine for dengue in the truest sense is urgently needed for its prevention.

It is still challenging to develop a dengue vaccine that offers complete protection against all four DENV serotypes with no or little ADE. The several tetravalent live-attenuated dengue vaccine designs and their substantial human evaluations have provided insight into vaccine design considerations. An efficient dengue vaccine should be within reach in the near future with the introduction of new technologies, such as mRNA-based vaccines.

## Figures and Tables

**Figure 1 vaccines-10-01940-f001:**
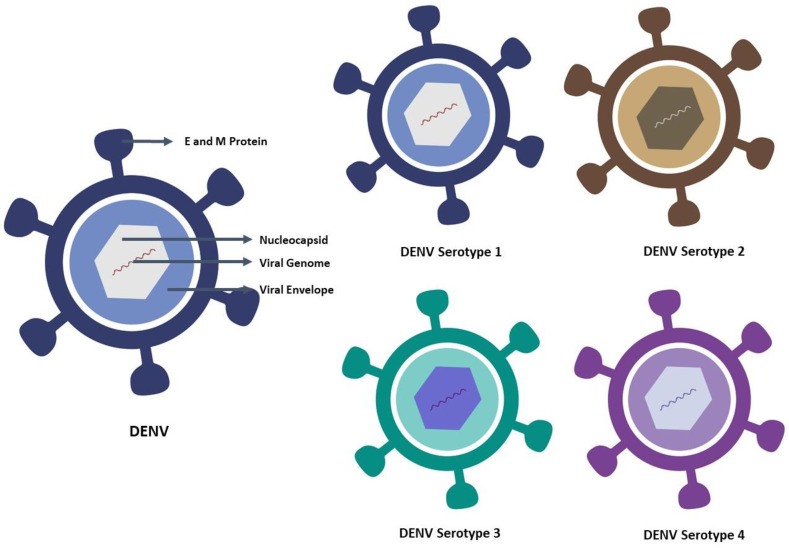
DENV and DENV serotype: The dengue virus is roughly spherical. Dengue infections are caused by four closely related viruses, named DENV-1, DENV-2, DENV-3, and DENV-4. An individual develops immunity to a specific dengue virus after recovering from an infection.

**Figure 2 vaccines-10-01940-f002:**
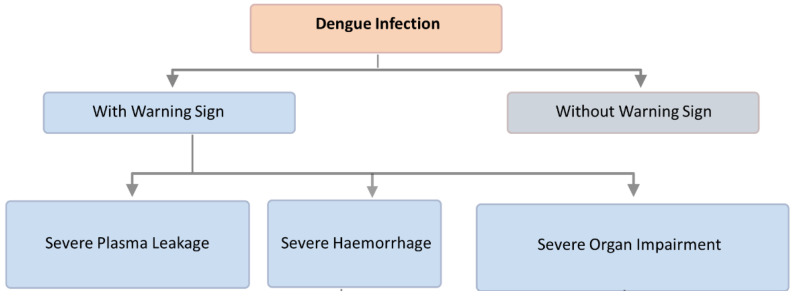
Classification of dengue disease: Dengue can range from an infection with no symptoms to a serious condition, as shown in the flowchart of Figure 2 (Note: classification based on WHO dengue guidelines—2009).

**Figure 3 vaccines-10-01940-f003:**
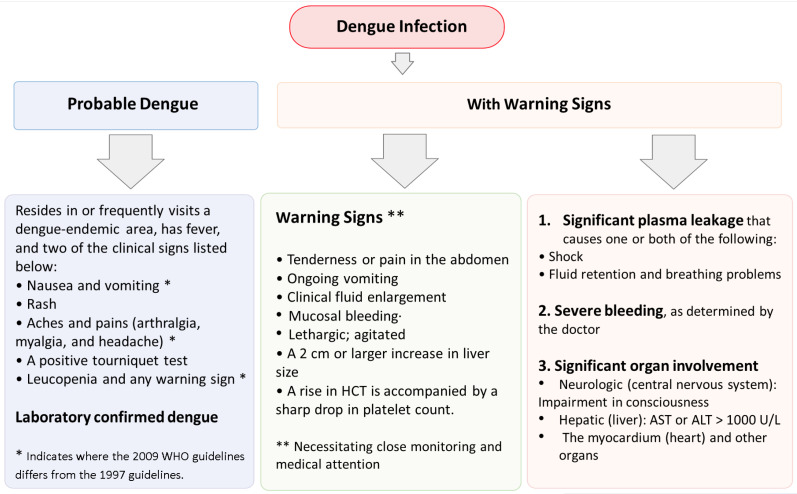
Clinical presentation of dengue infection and features: The clinical signs of dengue are outlined in the figure, as per classification based on WHO dengue guideline—2009. (Adapted from Dengue, Guidelines for Diagnosis, Treatment, Prevention and Control, WHO/TDR).

**Figure 4 vaccines-10-01940-f004:**
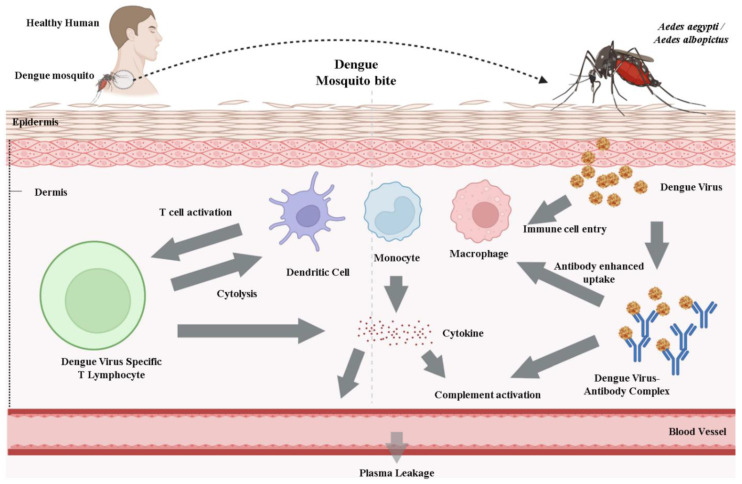
Pathogenesis of DENV: Released viral particles by mosquito bite may cause local immune cells to activate, primarily monocytes or dendritic cells (DCs). Natural killer (NK) cells and T cells are recruited because of a local inflammatory reaction to DENV in the skin, which enhances the death of virus-infected cells at the injection site. DENV will spread along the lymphatic pathways to draining lymph nodes, resulting in systemic infection.

**Figure 5 vaccines-10-01940-f005:**
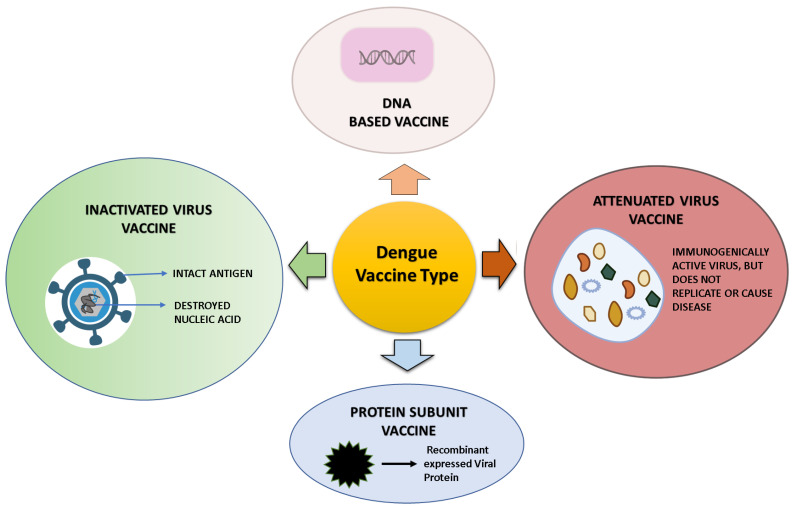
Dengue vaccine platform: Vaccines against the dengue virus have been developed using a range of platforms and funded by private companies as well as government agencies.

**Figure 6 vaccines-10-01940-f006:**
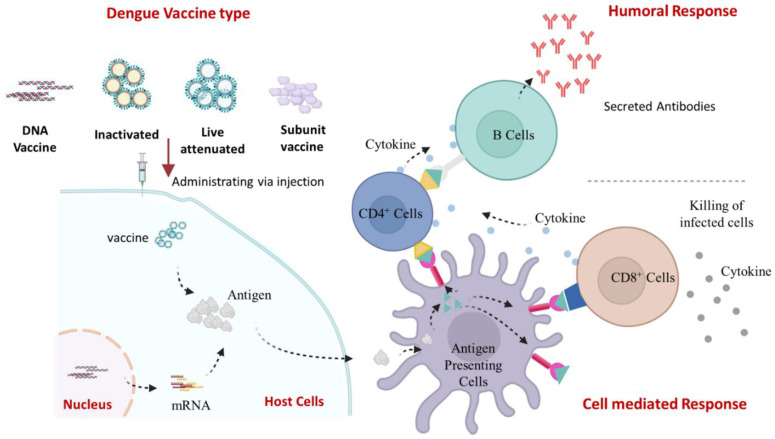
Therapeutic mechanism of a dengue vaccine: Vaccine was administered subcutaneously, and antigen-presenting cells (APC) present the processed antigen, which stimulates T lymphocytes. T lymphocytes promote the development of B cells. The maturation of the antibody response occurs because of B-cell proliferation, which is dependent on the T cells. Neutralizing antibodies (Nabs) specific for the dengue vaccine are secreted preferentially by plasma cells. Activated CD8 T cells can mount a response against infection by secreting cytokines and killing off viral infected cells.

**Table 1 vaccines-10-01940-t001:** Diagnostic methods for the detection of dengue infection, as adapted from the World Health Organization [45].

	Clinical Sample	Approach	Technique	Time Required
Detection of virus and viral product	Acute serum (1–5 days after onset of fever) and necropsy tissue	Isolation of virus Detection of nucleic acid Detection of antigen	Mosquito or mosquito cell culture	1 week or more
			RT-PCR and real-time RT-PCR	1–2 days
			NS1 antigen rapid test	Minutes
			NS1 antigen-capture ELISA	1 day
			Immunohistochemistry	2–5 days
Serological response	Paired sera S1: acute serum after 1–5 days S2: convalescent serum after 15–21 days Serum after 5 days of fever	Seroconversion (S1 to S2) of immunoglobulin M or immunoglobulin G Detection of immunoglobulin M Detection of immunoglobulin G	ELISA, HI	1–2 days
			Plaque reduction neutralization test	>7 days
			MAC-ELISA	1–2 days
			Immunoglobulin M rapid tests (LFA)	Minutes
			Immunoglobulin G ELISA, HI	1–2 days
			Immunoglobulin G rapid tests (LFA)	Minutes

Abbreviations: ELISA: Enzyme-linked immunosorbent assay; HI: Hemagglutination inhibition assay; MAC: Immunoglobulin M antibody capture; NS1: Non-structural protein 1; LFA: Lateral flow assay; RT-PCR: Reverse-transcription polymerase chain reaction.

**Table 2 vaccines-10-01940-t002:** Present status of different vaccine competitors based on live-attenuated/purified inactivated virus/recombinant subunit/nucleic acid (DNA) for dengue.

**Vaccine Candidate, Developer, and Status**	**Approach and Type of Vaccine**	**Status**	**References**
Dengvaxia ^®^ (CYD-TDV); Sanofi Pasteur; Licensed	YF-17D backbone with prM and E genes of dengue virus 1–4 (Live-attenuated vaccine)	Age limit; increased risk of severe dengue in seronegative subjects but high effectiveness and safe in seropositive individuals	[87,94]
TetraVax-DV-V003/TV005; NIAID/NIH; Phase III	Full-length dengue virus 1/2/3/4 lacking 30 nucleotides in 3′ UTR (Live-attenuated vaccine)	Well tolerated; balanced immune response in subjects, effective with administration of a single dose; adverse reaction (mild rash)	[88,89,95]
DENVax/TAK003/TDV; Takeda/Inviragen; Phase III	Dengue virus 2 PDK53 backbone with dengue virus 1,3, and 4 prM and E gene chimera (Live-attenuated vaccine)	Immunogenic and well tolerated in multiple phase I and II clinical studies, independent of the participant’s age or serostatus, safety profile not entirely known	[90,96]
TDEN LAV; WRAIR/GSK; Phase II (Discontinued)	Dengue virus 1/2/3/4 PDK virus (Live-attenuated vaccine)	Proven to be a safe, well-tolerated, and immunogenic DENV vaccine candidate in phase II trial	[91,97,98]
TDEN PIV; WRAIR/GSK/Fiocruz; Phase I	Formalin inactivated dengue virus 1/2/3/4 (Purified inactivated vaccine)	Well tolerated, immunogenic in naïve and seropositive individuals. No risk of re-activation and good immunological balance	[92,98]
V180; WRAIR/GSK/Fiocruz; Phase I/II	DEN-80E-containing recombinant truncated protein (Recombinant subunit)	Induced steady immune response against all DENV serotypes, decreasing the likelihood of the ADE effect	[99,100]
TVDV; US Naval Medical Research Centre; Phase I	prM and E proteins Nucleic acid (DNA)	Stable but lack of immunogenicity, Plasmid modification required	[101,102]
D1ME100; US Naval Medical Research Centre; Phase I	Recombinant plasmid vector encoding prM/E Nucleic acid (DNA)	No neutralizing antibody, Response detected in individuals with low-dose immunization	[103]

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
