# Peer review of "Preparedness for the Dengue Epidemic: Vaccine as a Viable Approach"

_vaccines, 2022, doi:10.3390/vaccines10111940_

Round 1

Reviewer 1 Report

The review article "Preparedness for the Dengue Epidemic: Vaccine a Viable Approach" by Md, Zeyaullah et. al. described the curent vaccines which have been in use or under clinical trial in various phrase. The authors did not discuss or provide an idea whether each vaccine format should be work well in the clinical trial. The authors can use the information of each vaccine format developed by invemtors to propose the advantage or disvantage of each vaccine should be in future use. 

Minor point:

1. There is some mistake in spelling of some text in Fig.1 and Fig.2 . The "Dengue shock symptom" should be Dengue shock syndrome in Fig.1, while "Fluid accumulating with reparatory distress" should be .......respiratory distress.

2. The wording plasmid leak syndrome in Fig.1 should be used other word because this is not familiar in dengue research.

Author Response

Review Report (Reviewer 1)

Comments and Suggestions for Authors:

The review article "Preparedness for the Dengue Epidemic: Vaccine a Viable Approach" by Md, Zeyaullah et. al. described the current vaccines which have been in use or under clinical trial in various phases. The authors did not discuss or provide an idea whether each vaccine format should be work well in the clinical trial. The authors can use the information of each vaccine format developed by inventors to propose the advantage or disadvantage of each vaccine should be in future use. 

Response: Each vaccine format developed by inventors was added in the text file with the red mark. In addition to that, Table 2 provide the status of the ongoing vaccine, and table 3 include the vaccine efficacy, serotype-specific efficacy, immunogenicity, and any side effect in the trial. A details description of each vaccine was given in the main body text in their respective heading.

Minor point:

  1. There is some mistake in spelling of some text in Fig.1 and Fig.2 . The "Dengue shock symptom" should be Dengue shock syndrome in Fig.1, while "Fluid accumulating with reparatory distress" should be .......respiratory distress.

Response: Thanks for pointing out the mistake. We have revised the figure based on revised DENGUE GUIDELINES FOR DIAGNOSIS, TREATMENT, PREVENTION, AND CONTROL. 2009 edition as suggested by another reviewer.

  1. The wording plasmid leak syndrome in Fig.1 should be used in other words because this is not familiar in dengue research.

Response: We also revised this figure based on revised DENGUE GUIDELINES FOR DIAGNOSIS, TREATMENT, PREVENTION, AND CONTROL. 2009 edition.

Reviewer 2 Report

In the following manuscript entitled “Preparedness for the Dengue Epidemic: Vaccine a Viable Approach” by Zeyaullah and colleagues, the authors highlighted the current trends on dengue vaccine candidates as well as the clinical results of their trials. Although there are several candidates for preventing dengue virus infection, neither effective nor public vaccines are available for safe application in endemic areas.  Thus, a better understanding of the new insights into their development and potential side effects of vaccine candidates represents an interesting topic that needs to be constantly discussed and updated to facilitate future public health interventions.  This is a well-written review; however, it needs to improve the organization of the depicted information in the manuscript, so readers could better follow up the main goal of this review.

Major(s) and minor(s):

1.       In the introduction section lines 38-39, the authors stated “…may result in dengue fever (DF) or a 38 lot of severe outcomes, like dengue hemorrhagic syndrome/shock”. Even though hemorrhage and shock syndrome, mainly associated with plasma leakage, is part of the severe manifestations occurring during severe dengue infections, these can occur one without the other and yet lead to severe dengue disease. Please separate these as indicated.

2.       Just a recommendation, I suggest switching the heading title “2. Incidence/Prevalence of Dengue Disease” to the heading number “3. Characteristic features of Dengue virus”.  

3.       In line 84, the authors stated that “DENV could be a mosquito-borne virus”. Please remove or change “could”. DENV is a mosquito-borne virus, or not?

4.       Also, taxonomically speaking, the virus genus and family should be written in italics. Please see lines 84 and 85. Same with mosquito species such as Aedes spp.

5.       Additionally, the author should decide in using either the abbreviated form for dengue virus “DENV” or keep using “dengue virus”. Please revise it.

6.       As pointed out above, dengue hemorrhagic fever (DHF) and Dengue shock Syndrome (DSS) are two separate entities that may occur together or separately during severe DENV infections. Please revise Figure 1. Please revise Figure 1.4 Suggested dengue case classification and levels of severity of the “DENGUE GUIDELINES FOR DIAGNOSIS, TREATMENT, PREVENTION, AND CONTROL. 2009 edition.

7.       Same with Figure 2. Please revise it based on what is described in the dengue guidelines for dengue case classification and levels of severity.

8.       How is Figure 3 about “Therapeutic mechanism of Dengue Vaccine” so ahead in the document? Can the author move it to later?

9.       In the same line, no immune response mechanisms triggered by DENV infection seem to be thoroughly discussed in this review. This must be included as it is a critical topic for vaccine development particularly for dengue disease.

10.   Additionally, how is the Dengue vaccine type (I, II, III, IV) related to any text included in the manuscript? Can the author clarify this?

Author Response

Review Report (Reviewer 2)

Comments and Suggestions for Authors

In the following manuscript entitled “Preparedness for the Dengue Epidemic: Vaccine a Viable Approach” by Zeyaullah and colleagues, the authors highlighted the current trends on dengue vaccine candidates as well as the clinical results of their trials. Although there are several candidates for preventing dengue virus infection, neither effective nor public vaccines are available for safe application in endemic areas.  Thus, a better understanding of the new insights into their development and potential side effects of vaccine candidates represents an interesting topic that needs to be constantly discussed and updated to facilitate future public health interventions.  This is a well-written review; however, it needs to improve the organization of the depicted information in the manuscript, so readers could better follow up the main goal of this review.

Response: Thanks for the suggestion. We reorganized the information in the manuscript and highlighted it with a red colour.

Major(s) and minor(s):

  1. In the introduction section lines 38-39, the authors stated “…may result in dengue fever (DF) or a lot of severe outcomes, like dengue hemorrhagic syndrome/shock”. Even though hemorrhage and shock syndrome, mainly associated with plasma leakage, is part of the severe manifestations occurring during severe dengue infections, these can occur one without the other and yet lead to severe dengue disease. Please separate these as indicated.

Response: Thanks for pointing out the mistake. We have corrected the main text part and highlighted with the red colour.

  1. Just a recommendation, I suggest switching the heading title “2. Incidence/Prevalence of Dengue Disease” to the heading number “3. Characteristic features of Dengue virus”.  

Thank you for your suggestion. The header has been rearranged within the body content and is now highlighted in red.

  1. In line 84, the authors stated that “DENV could be a mosquito-borne virus”. Please remove or change “could”. DENV is a mosquito-borne virus, or not?

Response: Thanks for pointing out the mistake. We have corrected the main text part with a red highlight.

  1. Also, taxonomically speaking, the virus genus and family should be written in italics. Please see lines 84 and 85. Same with mosquito species such as Aedes spp.

Response: Thanks for pointing out the mistake. We have corrected in the main text part and highlighted with the red colour.

  1. Additionally, the author should decide in using either the abbreviated form for dengue virus “DENV” or keep using “dengue virus”. Please revise it.

Response: Thanks for pointing out the mistake. We used the DENV in the text body.

  1. As pointed out above, dengue hemorrhagic fever (DHF) and Dengue shock Syndrome (DSS) are two separate entities that may occur together or separately during severe DENV infections. Please revise Figure 1. Please revise Figure 1.4 Suggested dengue case classification and levels of severity of the “DENGUE GUIDELINES FOR DIAGNOSIS, TREATMENT, PREVENTION, AND CONTROL. 2009 edition.

Response: Thanks for pointing out the mistake. We have revised the figure as per your suggestion.

  1. Same with Figure 2. Please revise it based on what is described in the dengue guidelines for dengue case classification and levels of severity.

Response: We also revised figure 2 as per suggestion.

  1. How is Figure 3 about “Therapeutic mechanism of Dengue Vaccine” so ahead in the document? Can the author move it to later?

Response: Thanks for the suggestion. We placed this content lately in the manuscript.

  1. In the same line, no immune response mechanisms triggered by DENV infection seem to be thoroughly discussed in this review. This must be included as it is a critical topic for vaccine development particularly for dengue disease.

Response: Thanks for the suggestion. We included this section in the revised manuscript with the figure.

  1. Additionally, how is the Dengue vaccine type (I, II, III, IV) related to any text included in the manuscript? Can the author clarify this?

Response: Thanks for pointing that out. A details description of each vaccine was given in the main body text in their respective heading.

Round 2

Reviewer 1 Report

The authors has been revised all information that was suggested from reviewers. There were much be clearer in the point of comments.

Author Response

The language was improved using professional english editor.  

Reviewer 2 Report

Just one minor but important observation that needs to be addressed by the authors as follows:

1. In Figure 1, Flaviviruses such as DENV are single-positive stranded RNA viruses. In Figure 1, the viral genome looks more like a double-strand viral genome. Can the authors modify this, so it looks more like a single-stranded RNA molecule?

Author Response

Figure 1 was modified as per reviewer suggestion. English language was improved and marked in the manuscript.